# Effect of Ultrasonication on the Morphology, Mechanical Property, Ionic Conductivity, and Flame Retardancy of PEO-LiCF_3_SO_3_-Halloysite Nanotube Composites for Use as Solid Polymer Electrolyte

**DOI:** 10.3390/polym14183710

**Published:** 2022-09-06

**Authors:** Pattranuch Pongsuk, Jantrawan Pumchusak

**Affiliations:** 1Department of Industrial Chemistry, Faculty of Science, Chiang Mai University, 239 Huay Kaew, Su Thep, Mueang, Chiang Mai 50200, Thailand; 2Materials Science Research Center, Faculty of Science, Chiang Mai University, 239 Huay Kaew, Su Thep, Mueang, Chiang Mai 50200, Thailand

**Keywords:** solid polymer electrolyte, halloysite nanotube, ultrasonic, polymer composite, poly (ethylene oxide), battery, clean energy

## Abstract

PEO-LiCF_3_SO_3_-halloysite nanotube (HNT) composites were fabricated by solution casting together with hot compression to form a solid polymer electrolyte (SPE) membrane. Different ultrasonic exposure times were used to disperse HNT nanoparticles in the PEO-20%LiCF_3_SO_3_-HNT composite solutions prior to casting. An exposure time of 15 min gave the highest ionic conductivity in the SPE membrane, the ionic conductivity significantly increased by two orders of magnitude from 6.6 × 10^−6^ to 1.1 × 10^−4^ S/cm. TEM, FE-SEM, and EDS-mapping were used to study the dispersion of HNTs in the SPE membrane. ATR-FTIR revealed that the bonding of PEO-LiCF_3_SO_3_ and PEO-HNT was created. XRD and DSC showed a reduction in the crystallinity of PEO due to HNT addition. The ultrasonication for an optimal period gave uniform dispersion of HNT, reduced the polymer crystallinity and strengthened the tensile property of SPE membrane. Moreover, the electrochemical stability, flame retardance and dimensional stability were improved by the addition of HNT and by ultrasonication.

## 1. Introduction

Currently, lithium-ion batteries (LIBs) typically contain liquid electrolytes (LEs), which limit operation to ≤4.8 V [1]. LEs are organic liquids or gels the use of which present safety issues due to risks such as generation of volatile gaseous species, leakage, thermal runaway and flammability [2]. Safer high-performance non-flammable electrolytes and solid electrolytes are needed to overcome such risks. Solid electrolytes, consisting of complex Li inorganic compounds or polymer composites, can also inhibit the growth of Li dendrites, can operate at higher voltages and be charged at higher rates, and their use may enable reductions in battery size. Solid polymer electrolytes (SPEs) have attracted increasing attention as potential solid electrolyte materials because of their flexibility in fabrication, chemical stability (non-reactive to Li metal), and wide electrochemical stability [3,4]. Considerable research efforts have been aimed at the technical and commercial development of SPE for solid-state LIBs. Electric vehicles and mobile phones are expected to benefit from solid-state LIBs because of their higher safety during practical use. However, there is a still a need for further SPE development and improvement since current SPE remain limited by poor ionic conductivity and poor mechanical properties. An ionic conductivity higher than 10^−4^ S/cm is the target for SPE to be considered for practical use in LIBs [5]. Poly (ethylene oxide) (PEO) is usually chosen as a matrix for SPE because lithium salts are well soluble in PEO [6], thus facilitating generation of charge carriers for conducting ions. However, the PEO matrix shows lower ionic conductivity due to the presence of crystalline domains in its structure at room temperature [7]. The mixture of PEO and lithium salt provides ionic conductivity lower than 10^−6^ S/cm. It has been reported that poor ionic conductivity of SPEs is usually found when high crystallinity polymer matrices are used. Low ion conductivity is also caused if ion aggregation or anion pair formation occur in the system [8,9]. Therefore, to increase ionic conductivity in an SPE system, polymer crystallization must be inhibited. In addition, ion aggregation and anion pair formation should be avoided. Plasticizer has been incorporated into SPE to reduce polymer crystallinity in an effort to increase ionic conductivity; however, this usually reduces mechanical properties and lowers the electrochemical potential of SPE [7]. An interesting method to boost the ionic conductivity of SPE is the addition of inorganic fillers, which also offers the possibility to improve the mechanical properties of SPE [8,9]. However, composite-based SPE that contains ceramic fillers suffer from the problem of particle aggregation at high content which reduces the number of dissociated Li^+^ ions and decreases ionic conductivity [10]. Ultrasonication has been widely used to disperse nanoparticles in polymer matrices [11,12]. However, excessive ultrasonic exposure times may cause a re-agglomeration of nanoparticles [13] or degrade the properties of nanocomposite due to an increased temperature [12]. Jokar et al. [14] showed that ultrasonication provided ease of dispersion of clay in a polymer matrix, which led to an increase of d-spacing of the clay structure. Parveen et al. [12] found that microcrystalline cellulose (MCM) had slightly decreased agglomeration when ultrasonication times of up to 30 min were applied; however, agglomeration increased at longer ultrasonication durations.

This work has selected halloysite nanotubes (HNTs) for use as nanofiller because of their large surface area, high porosity, and tunable surface chemistry [15]. The chemical structure of HNT consists of silicon dioxide tetrahedrons at the outer surface of the tube and aluminum oxide octahedrons at the inner surface of the tube with a lumen at the center. These characteristics should facilitate ion transfer for ion conductive materials [16,17]. Ultrasonic stirring was applied to facilitate the dispersion of HNTs in a PEO-20%LiCF_3_SO_3_-HNT composite. The effects of different ultrasonic exposure times on ionic conductivity, crystallinity, thermal properties, electrochemical stability, flame retardance and dimensional stability and mechanical properties of SPE were investigated.

## 2. Materials and Methods

### 2.1. Materials

PEO (M.W. 6 × 10^6^), lithium trifluoromethanesulfonate (LiCF_3_SO_3_) and halloysite nanotube (HNTs; Al_2_Si_2_O_5_(OH)_4_.2H_2_O) were purchased from Sigma-Aldrich. Tetrahydrofuran (THF) was purchased from RCI Labscan.

### 2.2. Fabrication of Solid Polymer Electrolyte (SPEs) Nanocomposites

PEO and LiCF_3_SO_3_ were used to prepare SPEs as the base composites. HNTs were used as an inorganic nanofiller. The optimum concentration of LiCF_3_SO_3_ in PEO was investigated in previous work [16] which found that 20 wt.% of the salt gave the highest ionic conductivity. Therefore, this concentration was used to study the effect of HNT addition. Different amounts of HNT were added into PEO-20%LiCF_3_SO_3_ solution to give HNT contents of 5, 10 and 15 wt%. THF was used as the solvent for all composite solutions. A conventional mixing method using a magnetic stirrer was applied to stir each solution at room temperature for 24 h prior to casting to produce membrane. The membrane was dried in a vacuum oven at 60 °C for 2 h and then subjected to compression molding at 1500 psi for 10 min to form a thin SPE membrane. These samples were denoted as Non-US. The solution after magnetic stirring was sonicated using a high-intensity ultrasonic probe at 50% amplitude for 10 s at pulse mode (Sonics & Materials, Inc, Newtown, CT, USA, VCX750). The effect of ultrasonication was explored with various ultrasonic exposure times of 5, 10, 15, 30, and 60 min. SPE membranes were fabricated using the same route as for the Non-US material as outlined above. The ultrasonication SPE samples are denoted as US. The two fabrication routes are shown in Figure 1.

### 2.3. Measurements

The ionic conductivity was obtained by impedance spectra measurements using Autolab PGSTAT (302N, Metrohm Autolab Inc., Utrecht, Netherland) in the frequency ranges of 1 Hz–100 kHz at room temperature. The SPE membrane was inserted between two stainless steel (SS) blocking electrodes. An electric field was applied to the cell to cause movement of Li^+^ ions. The bulk resistance (R_b_) of the SPE was calculated from the intercept of the *x*-axis on a Nyquist plot at high frequency. The ionic conductivity of the SPE was calculated using the Equation (1),
σ = t/R_b_·A(1)
where t is the thickness, R_b_ is the bulk resistance, and A is the area of the electrolyte membrane.

The electrochemical stability windows of SPE were determined by linear sweep voltammetry (LSV) that was performed on SS/SPE/SS cells at room temperature. The LSV was measured from 0 to 5 V at a scan rate of 5 mV s^−1^. DSC measurements were performed by a DSC1 calorimeter (Mettler Toledo). The sample was subjected to a temperature scanning range from −70 °C to 110 °C at a heating rate of 10 °C/min. Values for glass transition temperature (T_g_), melting point (T_m_), and % crystallinity were obtained from this experiment. The X-ray diffraction spectra of the SPE membranes were determined using a Rigaku rotating anode goniometer (Rigaku Denki., Ltd., Tokyo, Japan). XRD scans were obtained using Cu Kα radiation (1.541 °A) operating at 40 kV and 100 mA in the 2θ range of 10° to 60° using a step width of 0.02°. FTIR spectra were obtained using an ATR- FTIR Perkin Elmer (Frontier L128-0010) within a range of 450–4000 cm^−1^ and using a resolution of 4 cm^−1^. A field-emission scanning electron microscope (FE-SEM, JSM 6335 F) equipped with an energy dispersive X-ray (EDX) analyzer operated at 15 kV was utilized to monitor the form of the HNT dispersion in SPE. Transmission electron microscopy (TEM, TALOS F200X, Thermo Fisher Scientific, Bleiswijk, Netherlands) was used to observe the distribution and the length of HNT particles in the SPE membranes. The tensile properties of the SPE were determined using a Lloyd universal testing machine (model LRX). The samples were tested using a crosshead speed of 50 mm/min at room temperature. The size of each test piece size was 100 mm (length) × 5mm (width) × 0.1–0.2 mm (thickness), with a gauge length of 50 mm. Flame retardant testing was performed on the sample membranes (2 × 3 cm^2^). The distance between sample membrane and the top of the flame was 2 cm. The time required to start ignition and the time for self-extinguishing after removal of the ignition source were recorded. The dimensional stability of the sample membrane (2 × 3 cm^2^) was evaluated by observing the dimensional changes after the membrane was held for 5 and 10 h at 60 °C in an electric oven.

## 3. Results

### 3.1. Effect of Ultrasonication on the Dispersion of HNTs in SPE

#### Morphology Study

The incorporation of nanoparticles (NPs) in a polymer matrix is usually limited by the problem of aggregation of the NPs, thus reducing their reinforcing capability. In this work ultrasonication was applied to PEO-20%LiCF_3_SO_3_-HNT mixtures to improve the dispersion of the HNTs. The effects of ultrasonic exposure times were investigated at 5, 10, 15, 30, and 60 min and the resultant HNT dispersions compared to those from the conventional magnetic stirrer method. SEM and TEM were used to investigate the morphology of the SPE membranes and the dispersion of HNT in the PEO matrix. The HNTs exhibited a typical nanotube structure (Figure 2a). The nanotubes had an external diameter of 50–100 nm, an inner diameter of 6–30 nm, and are 60–480 nm in length [18]. The tubular nature of the HNT was confirmed by XRD spectra, which show a high-intensity peak at 12.03° (Figure 2b). SEM-EDS was used to inspect the distribution of HNTs in the PEO matrix. Figure 2c shows the distributions of Al, Si, and O atoms which correspond with the alumina and silica in the HNT. For the Non-US treated mixes, the SEM image in Figure 3 shows that the composite with 10 wt.% HNT is more prone to HNT aggregation formation compared with the composite with 5 wt.%. The SEM-EDS mapping of PEO-20%LiCF_3_SO_3_-5%HNT shows the relatively uniform distribution of Al (red) and Si (green), indicating that the HNT was dispersed homogenously in the SPE.

Ultrasonication was applied to the PEO-20%LiCF_3_SO_3_-5%HNT and the PEO-20%LiCF_3_SO_3_-10%HNT samples for various ultrasonic exposure times. The PEO-20%LiCF_3_SO_3_-5%HNT sample was studied in more detail to determine the effect of ultrasonication on the membrane properties. Figure 4 presents SEM images and EDS mapping for Si and Al to reveal the distribution of HNT in the PEO matrix for the 5% HNT addition. It was found that after 15 min of ultrasonication, the HNT appears to be well dispersed in the PEO matrix, as confirmed by the TEM image in Figure 5. Ultrasonication for 15 min. prevented formation of aggregates giving a uniform dispersion of nanoparticles in the polymer matrix. However, as illustrated by Figure 5c,d, when the ultrasonic exposure times were increased to 30 and 60 min, the lengths of the HNT tubes were reduced by fragmentation and re-agglomeration of the HNTs was observed in the PEO matrix. Measurements from TEM images revealed that the average length of HNT decreased from 482 ± 215 nm to 275 ± 108, 241 ± 56, and 216 ± 81 nm for the ultrasonic exposure times of 0, 15, 30, and 60 min, respectively. The TEM images (Figure 5) also provided information on the arrangement of the HNTs in the membrane matrix. Without ultrasonication, the HNT tubes were aligned in the matrix as indicated by the red arrows in Figure 5a. Using ultrasonication gave rise to random orientation of HNTs in the membrane as shown in Figure 5b–d. The physical and mechanical properties of polymer nanocomposites strongly depend on the orientation of nanoparticles [19]. It is well known that the reinforcement effect is greater when the fillers are aligned in the direction of strain (one-direction) but is much less in other directions. However, for application as SPE, random orientation is preferred. Mechanical properties in terms of the tensile behavior of the composites produced in this study and the effects of ultrasonication are discussed later in Section 3.7.

### 3.2. Effect of Ultrasonication on Crystallinity and Thermal Properties of SPE

#### 3.2.1. XRD Analysis

The effect of ultrasonic exposure time (0, 15, 30, and 60 min) on the crystallinity change of PEO is shown by the XRD patterns in Figure 6. The XRD pattern of neat PEO contained two main diffraction peaks at 2θ = 19.14° (1 2 0) and 23.3° (0 3 2), (1 1 2) [20,21,22,23] corresponding to the crystalline phase of the PEO polymer matrix. The addition of the LiCF_3_SO_3_ salt reduced the crystallinity of the PEO, as indicated by the significant decrease in the intensity of the prominent peaks at 19.33° and 23.3°. The decrease in the crystallinity of the PEO matrix is due to the complexation between PEO and Li^+^ ions that inhibits crystallization of the PEO [24]. It was also found that the intensity of the main peaks of PEO decreased upon the incorporation of 5 wt.% HNT. The explanation for this crystallinity decrease is that the HNT may impede the diffusion of the polymer chain to the crystal growth front [25]. When 15 min of ultrasonication was applied to the composite solutions, the diffraction peaks of PEO in the composite became broader and lower in intensity (Figure 6 pattern (d)). The main peak, located at 19.00°, was displaced to a lower angle indicating an increase of d-spacing [26]. However, the second main peak became divided into multi-peaks. When the ultrasonic exposure times increased to 30 and 60 min, the intensity of the peaks tended to increase, which corresponded to increasing crystallinity. It was also found that the prominent peaks were displaced to higher angles, corresponding to decreases in d−spacing.

#### 3.2.2. DSC Analysis

It is well known that Li^+^ can move in two ways: (a) along the molecular chains of a polymer, and (b) in the amorphous phase of a polymer electrolyte [27]. Therefore, the ionic conductivity of SPE should relate to decreases of T_g_ and the proportion of crystalline phase of the SPE. T_g_ and % crystallinity of the SPE were evaluated by DSC. The resultant DSC chromatogram of SPE is shown in Figure 7. The endothermic peak of PEO−20%LiCF_3_SO_3_ at 58.2 °C is the melting peak of SPE. The addition of 5%HNT combined with 15 min ultrasonic exposure reduced the area of the melting peak and lowered T_g_ of the PEO, indicating that the fraction of the crystalline phase in the SPE was decreased. The crystallinity of PEO is calculated by Equation (2)
X_c_ = ΔH_m_/(W_i_ × ΔH°_m_)(2)
where X_c_ is crystallinity (%), Wi is the weight fraction of the polymer in the composite, ΔH_m_ is the melting enthalpy of a sample, and ΔH°_m_ is the melting enthalpy of reference for 100% crystallinity of the polymer matrix (PEO = 213.7 J/g) [28]. The T_g_ of PEO was reduced from −60.7 °C to −67.9 °C by the addition of 5%HNT. Decreases in T_g_ in the SPE mean that more flexible polymer chains are obtained [29]. The lowest degree of crystallinity of the SPE was found in PEO-20%LiCF_3_SO_3_−5%HNT−15 min. This result agreed well with XRD results. The crystalline fraction (X_c_) from the XRD pattern can be calculated by Equation (3)
X_c_ = A_c_/(A_c_ + A_a_)(3)
where A_c_ and A_a_ are the crystallized and amorphous areas. Decreases of T_g_ and crystallinity lead to increased ion transportation within the SPE [29]. Higher ionic conductivity is related to the lowest crystallinity, which in this work was found in the PEO−20%LiCF_3_SO_3_−5%HNT−15 min composite. When ultrasonication was applied for 30 min and 60 min, the crystallinity tended to increase since the longer time led to segmentation of the polymer chains. The resultant shorter chain lengths were then able to reorient easily to give the higher crystallinity [30]. This consequently restricted the mobility of the polymer chains, thus giving a decrease in ionic conductivity. The values of T_g_, T_m_, ΔH_m_, and X_C_ of the various samples are listed in Table 1.

### 3.3. Effect of Ultrasonication on PEO−LiCF_3_SO_3_−HNT Complex

The FT-IR technique has been used to investigate the complexation between Li^+^ and polymer matrices [31,32]. In SPE, the Li^+^ would transport upon the application of current into the electrolyte membrane by an ion hopping mechanism [33]. The ether oxygen group (C-O-C) of PEO is strongly electronegative, thus it can readily undergo complexation with Li^+^ ions [34]. It is noted that the characteristic peak of C-O-C showed a triplet stretching vibration, located at 1147, 1097, 1060 cm^−1^ as shown in box 4 of Figure 8a. The shift of the C-O-C peaks suggests coordination between Li^+^ ions and C-O-C bonds in SPE [30]. According to literature, the bonds located at 1032 cm^−1^, 1042 cm^−1^, and 1056 cm^−1^ are assigned to free ions, ion pairs, and ion clusters, respectively [35,36]. FT−IR spectra of PEO−20%LiCF_3_SO_3_ displayed peaks from both ion pairs and free ions, which were located at 1041, 1029 cm^−1^, respectively. The spectra shows that the addition of HNT and the use of ultrasonic treatment caused the disappearance of the peak related to ion pairs, which corresponds with the increase in the free ions peak located at 1041 cm^−1^. This observation indicates that the addition of HNTs and the use of ultrasonication can dissociate LiCF_3_SO_3_ in SPE by reducing the formation of ion pairs. On the other hand, an anion of a lithium salt (CF_3_SO_3_^−^, ionic radius is 0.270 nm) may react with aluminal (Al-OH) on the inner surface of HNT (diameter of the lumen is 20–40 nm) thus causing reduced ionic bonding strength between Li^+^ and CF_3_SO_3_^-^ [37,38]. This effect provided more dissociation of lithium salt in the polymer matrix, enhancing the ionic conductivity of the SPE. The schematic illustration of the Li^+^ conduction mechanism in PEO–LiCF_3_SO_3_−HNT is shown in Figure 8b. The shift of the vibration bond is listed in Table 2.

### 3.4. Effect of Ultrasonication on Ionic Conductivity

The Nyquist plots of all samples show only the spike (Figure 9a), which can be attributed to ionic diffusion. This signifies good ionic conductivity of the SPE membranes. The bulk resistance (R_b_) can be obtained from where the plot of the results intercepts the *x*-axis. The small value of R_b_ proves that the current easily transfers through the electrode–electrolyte interface due to the high mobility of Li^+^ [39]. The ionic conductivity was found to increase with the addition of HNTs. The highest ionic conductivity was found at 5 wt.% of HNT. This can be ascribed to increasing lithium salt dissociation due to the interaction between the oxygen atoms of PEO and Li^+^ ions on the external surface of HNT (−Si-O-Si- group) [6] and the formation of hydrogen bonds between anions (CF_3_SO_3_^−^) and -OH groups on the internal surfaces of HNT [40]. However, at 10 and 15 wt.% loadings of HNT into PEO−20%LiCF_3_SO_3_ the ionic conductivity decreased. This decrease in ionic conductivity may be caused by a blocking effect where the HNT particles are too close to one another [41]. The effect of HNT content on the ionic conductivity of SPE is presented in Figure 9b. The results also show that the application of ultrasonication gave higher ionic conductivity than by using only the conventional magnetic stirring method. The highest ionic conductivity, which corresponded to the lowest R_b_ value, was found in the PEO−20%LiCF_3_SO_3_−5%HNT sample that was obtained using 15 min of ultrasonication (Figure 9c). This indicates that ultrasonication improved the dispersion of HNTs in the PEO matrix. As a consequence, the Lewis acid–base interaction could readily take place thus enhancing the mobility of ions and increasing the ionic conductivity. When ultrasonic stirring was applied for 30 and 60 min, the ionic conductivity was reduced by about one order of magnitude (Table 3). This reduction was caused by HNT re-agglomeration that hindered the motion of the mobile ions. The same pattern of ionic conductivity results was found when ultrasonication was also applied to the PEO−20%LiCF_3_SO_3_−10%HNT system. (see Table 3).

### 3.5. Effect of Ultrasonication on Electrochemical Stability

An important factor for SPE is the electrochemical stability window (ESW) which refers to safety aspects during application of voltage to a battery [42]. The ESW was determined by linear sweep voltammogram (LVS), using stainless steel (SS) as the working and reference electrodes. The decomposition of SPE was investigated in the potential range of 0–5 V at room temperature, as shown in Figure 9d. The decomposition of the composites without HNT, PEO-20%LiCF_3_SO_3_-5%HNT, and as PEO−20%LiCF_3_SO_3_−5%HNT−15 min was observed at the 2.96, 3.11, and 3.20 Volts, respectively, indicating that the electrochemical stability of SPE was improved when HNT was present and when ultrasonication was also applied.

### 3.6. Flame Retardance Testing and Dimensional Stability

The results of flame retardance testing in air of PEO−20%LiCF_3_SO_3_−5%HNT and PEO−20%LiCF_3_SO_3_−5%HNT−15 US in air are shown in Figure 10. The tests revealed that the PEO−20%LiCF_3_SO_3_−5%HNT was ignited at 4s after flame exposure as shown in Figure 10a. The specimen was burned completely at 13 s. Figure 10b shows the flame retardance characteristic of the PEO−20%LiCF_3_SO_3_−5%HNT−15 US membrane, which melted at 10 s and then ignited after 30s followed by self-extinguishing at 40 s. These observations indicate that suitably uniform dispersion of HNT in the composite is crucial for flame retardance. PEO−20%LiCF_3_SO_3_−5%HNT−15 US showed excellent flame retardancy which was better than that of the PEO−20%LiCF_3_SO_3_−5%HNT. Figure 10c shows dimensional changes of the SPE at 60 °C for 0, 5 and 10 h. It was found that the PEO−20%LiCF_3_SO_3_−5%HNT showed clear changes in the membrane size, while the PEO−20%LiCF_3_SO_3_−5%HNT−15 US membrane showed improved dimensional stability with less change than that of the non-ultrasonically treated membrane.

### 3.7. Effect of Ultrasonication on Tensile Properties

The tensile stress–strain curves of the SPEs are shown in Figure 11. The tensile strength of PEO−20%LiCF_3_SO_3_ increased by about 65% with the addition of 5%HNT. This indicated that the nanotube acted as a reinforcing agent such that the applied load could transfer from the matrix to the nanotube [43]. The tensile strength of PEO−20%LiCF_3_SO_3_−5%HNT−US 15 min increased by about 116% when compared to that of PEO−20%LiCF_3_SO_3_. Ultrasonication provided improved nanoparticle distribution; hence, load transfer was more effective providing better mechanical properties of the polymer composites [44,45,46]. The onset of plastic deformation was investigated at a 2% offset strain, which is a significant property measure when using materials for engineering applications. The 2% offset of a strain of PEO−20%LiCF_3_SO_3_−5%HNT−15 min showed the highest value of all the prepared composites in this study as 1.53 MPa. The tensile properties of SPE are listed in Table 4.

## 4. Conclusions

PEO−20%LiCF_3_SO_3_−HNT composite membranes were prepared by solution casting, ultrasonication and compression molding as SPE for potential application in lithium-ion batteries. It was found that ultrasonication could disperse the HNT nanoparticles in the sample solutions effectively thus reducing the crystallinity of the PEO. A treatment time of 15 min provided membrane with high ionic conductivity, good electrochemical stability, a 2% higher offset strain stress and higher tensile strength. Ultrasonication exposure time affected the HNT length and the properties of the SPE. PEO−20%LiCF_3_SO_3_−5%HNT subjected to 15 min of ultrasonication was found to be the optimal condition for SPE preparation, providing a SPE membrane with not only the high tensile properties but also highest ionic conductivity of 1.1 ± 0.19 × 10^−4^ S/cm. This SPE membrane could be stable up to 3.2 Volts and also showed excellent flame retardancy and dimensional stability. The uniform dispersion of HNT during preparation was found to be a key factor in producing composite membrane with optimum properties. Composite membrane is considered to be a suitable candidate for possible use as SPE in a solid-state Li-ion battery.

## Figures and Tables

**Figure 1 polymers-14-03710-f001:**
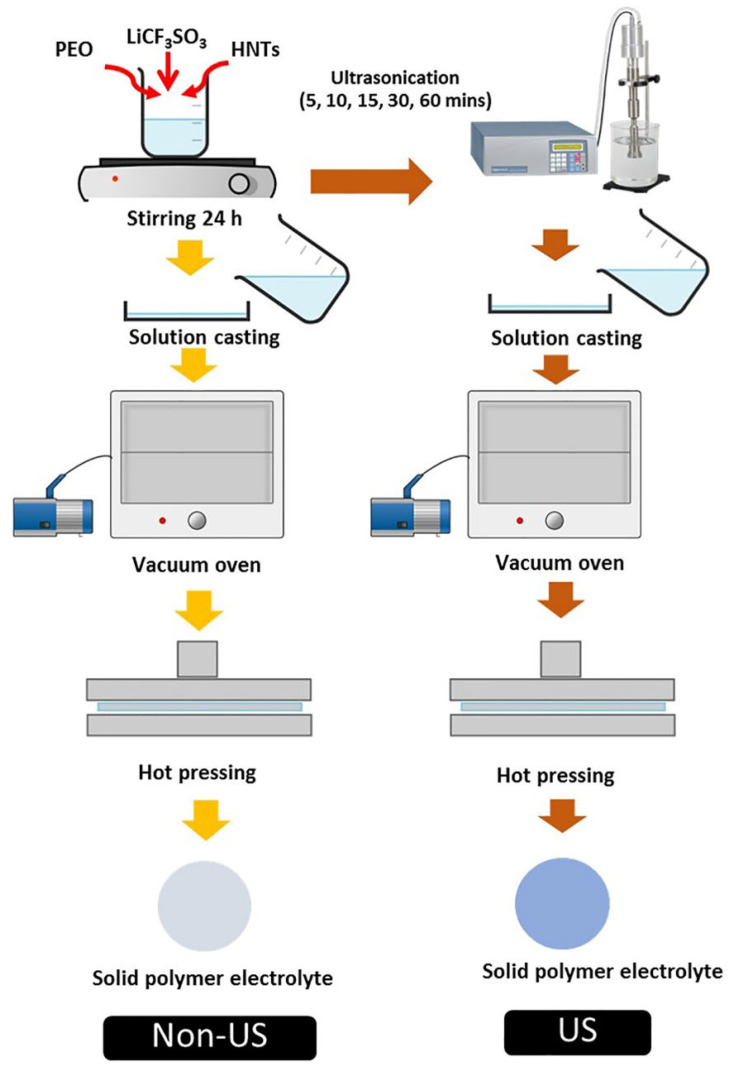
Schematic outline of the two fabrication routes for solid polymer electrolyte nanocomposite.

**Figure 2 polymers-14-03710-f002:**
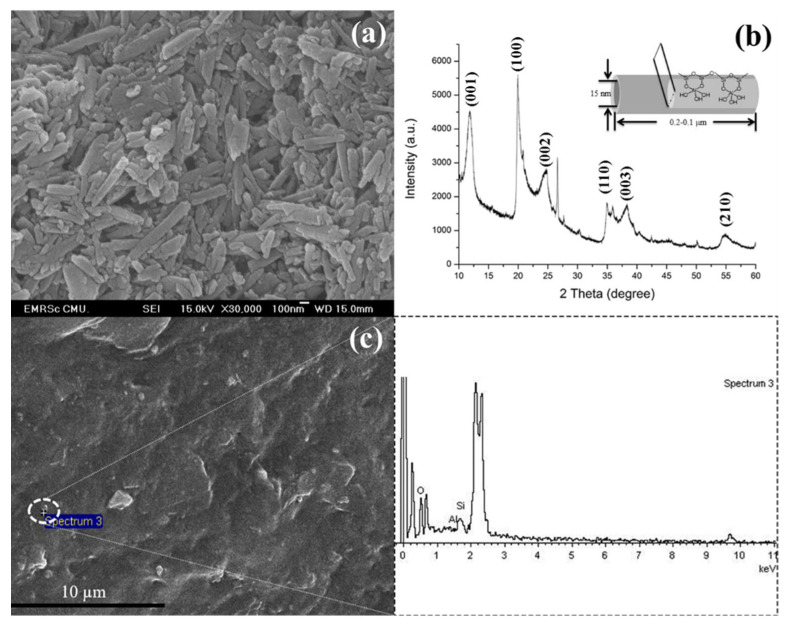
FE−SEM image of (**a**) As−supplied HNT, (**b**) XRD pattern and crystalline structure of HNT and (**c**) SEM image and EDS spectrum of HNT in SPE.

**Figure 3 polymers-14-03710-f003:**
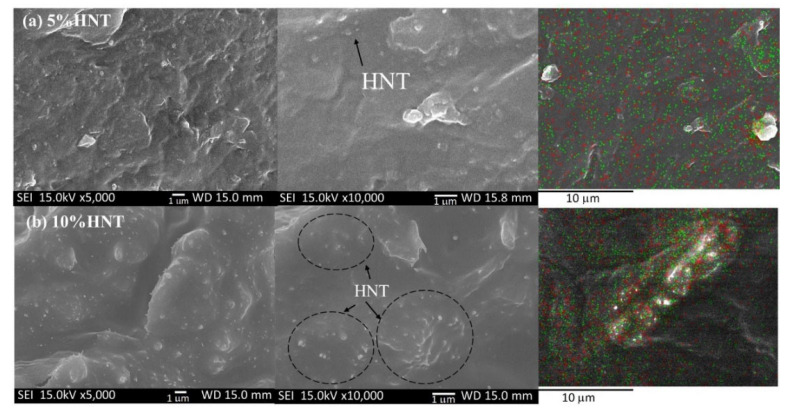
SEM micrographs of fracture surfaces of the Non−US prepared PEO-20%LiCF_3_SO_3_ with (**a**) 5%HNT, (**b**) 10%HNT at ×5000, ×10,000 and EDS mapping of Al (red), Si (green), respectively.

**Figure 4 polymers-14-03710-f004:**
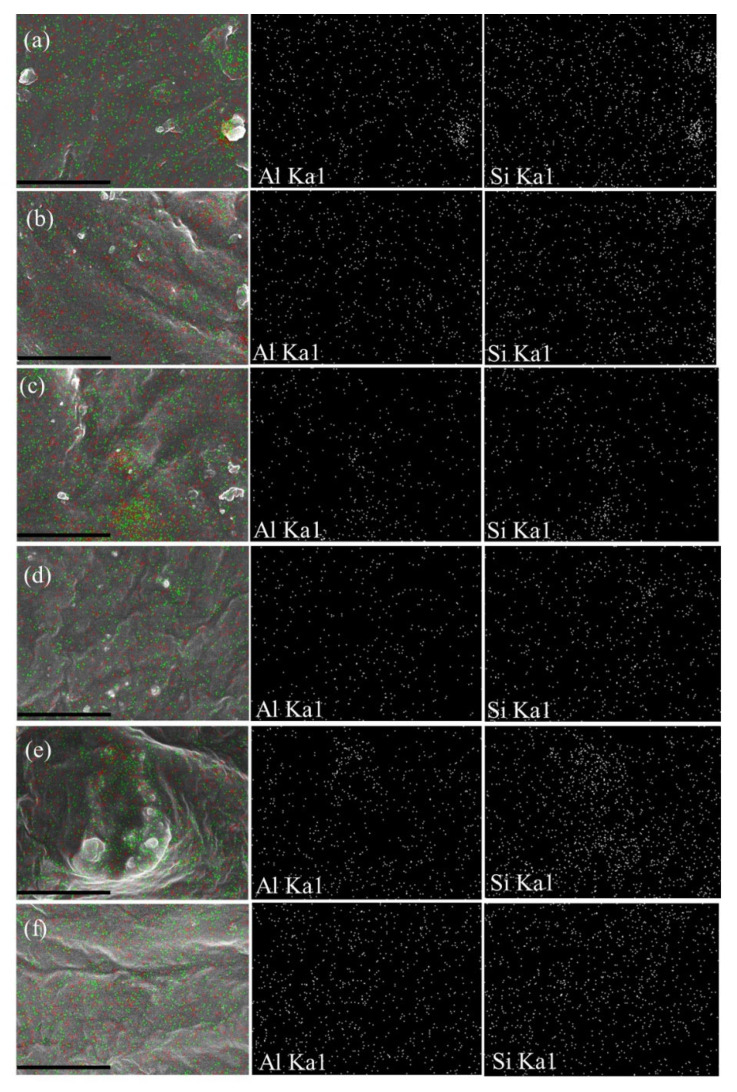
FE−SEM−EDS mapping at ×5000 (scale bar = 10 μm) of silicon (Si) atoms (green) and aluminum (Al) atoms (red) to indicate HNT distribution in PEO−20%LiCF_3_SO_3_−5%HNT of (**a**) Non−US, (**b**) US 5 min, (**c**) US 10 min, (**d**) US 15 min, (**e**) US 30 min and (**f**) US 60 min.

**Figure 5 polymers-14-03710-f005:**
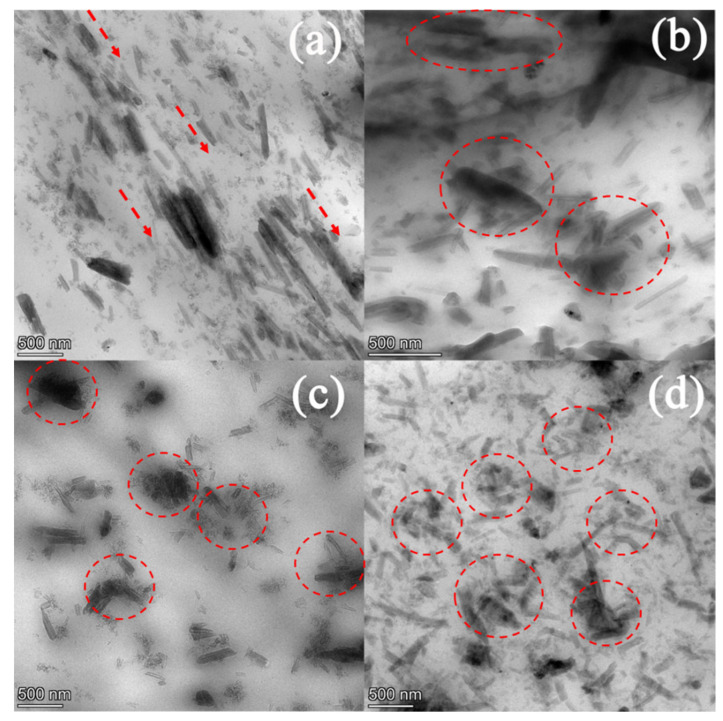
TEM images of PEO−20%LiCF_3_SO_3_−5%HNT (**a**) Non−US, (**b**) US 15 min, (**c**) US 30 min and (**d**) US 60 min.

**Figure 6 polymers-14-03710-f006:**
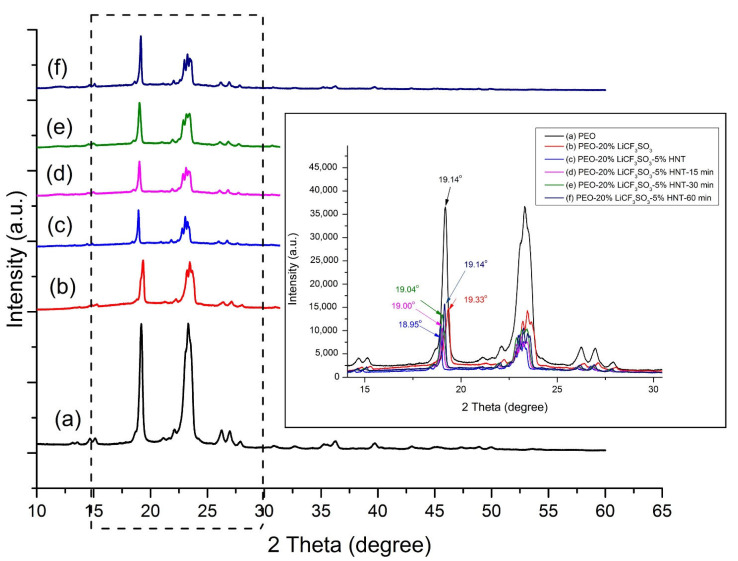
XRD patterns of (**a**) As−received PEO, (**b**) PEO−20%LiCF_3_SO_3_, (**c**) PEO−20%LiCF_3_SO_3_−5%HNT, (**d**) PEO−20%LiCF_3_SO_3_−5%HNT−15 min, (**e**) PEO−20%LiCF_3_SO_3_−5%HNT−30 min and (**f**) PEO−20%LiCF_3_SO_3_−5%HNT−60 min.

**Figure 7 polymers-14-03710-f007:**
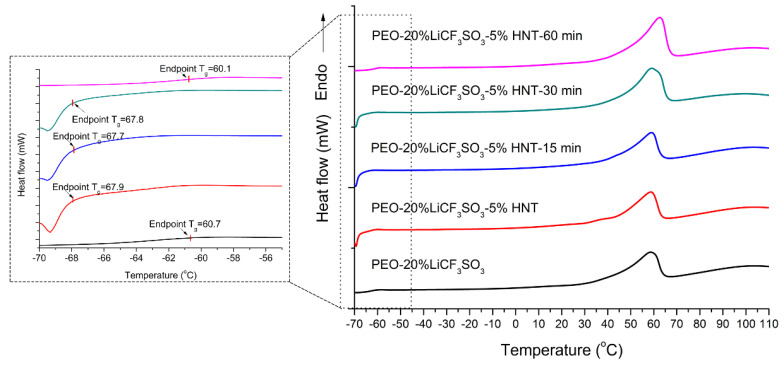
DSC curves of PEO−20%LiCF_3_SO_3_ and PEO−20%LiCF_3_SO_3_−5%HNT with and without ultrasonication.

**Figure 8 polymers-14-03710-f008:**
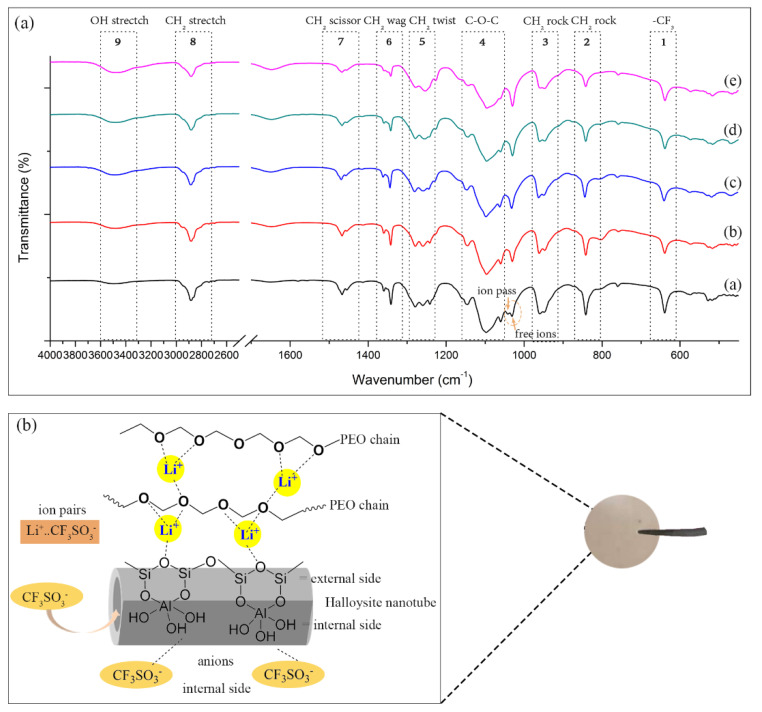
(**a**) ATR−FTIR spectra in the region of 450–4000 cm^−1^ of (a) PEO−20%LiCF_3_SO_3_, (b) PEO−20%LiCF_3_SO_3_−5%HNT, (c) PEO−20%LiCF_3_SO_3_−5%HNT−15 min, (d) PEO−20%LiCF_3_SO_3_−5%HNT−30 min, and (e) PEO−20%LiCF_3_SO_3_−5%HNT−60 min. (**b**) Schematic illustration of Li^+^ conduction mechanism in PEO−LiCF_3_SO_3_−HNT (left) and the sample of solid polymer electrolyte (right).

**Figure 9 polymers-14-03710-f009:**
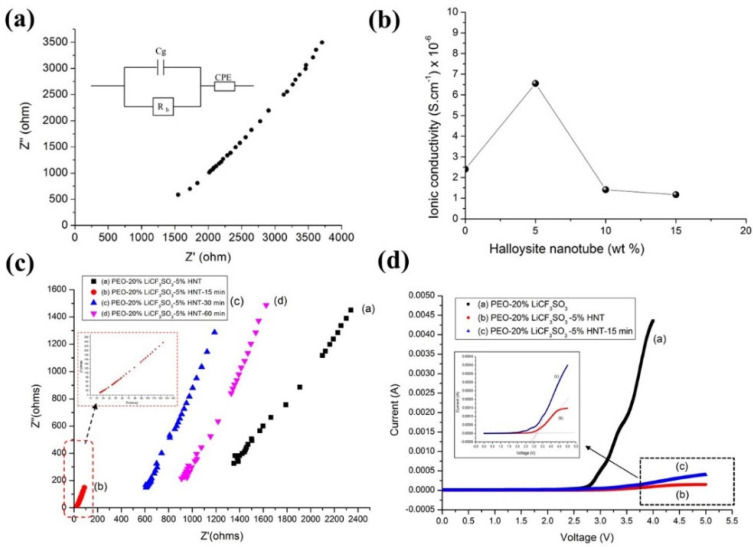
(**a**) Nyquist plot for Non−US of PEO−20%LiCF_3_SO_3_ at room temperature, (**b**) the ionic conductivity for Non−US of PEO−20%LiCF_3_SO_3_−y%HNT, (**c**) Nyquist plot of PEO−20%LiCF_3_SO_3_−5%HNT with various ultrasonic exposure time and (**d**) Linear sweep voltammogram.

**Figure 10 polymers-14-03710-f010:**
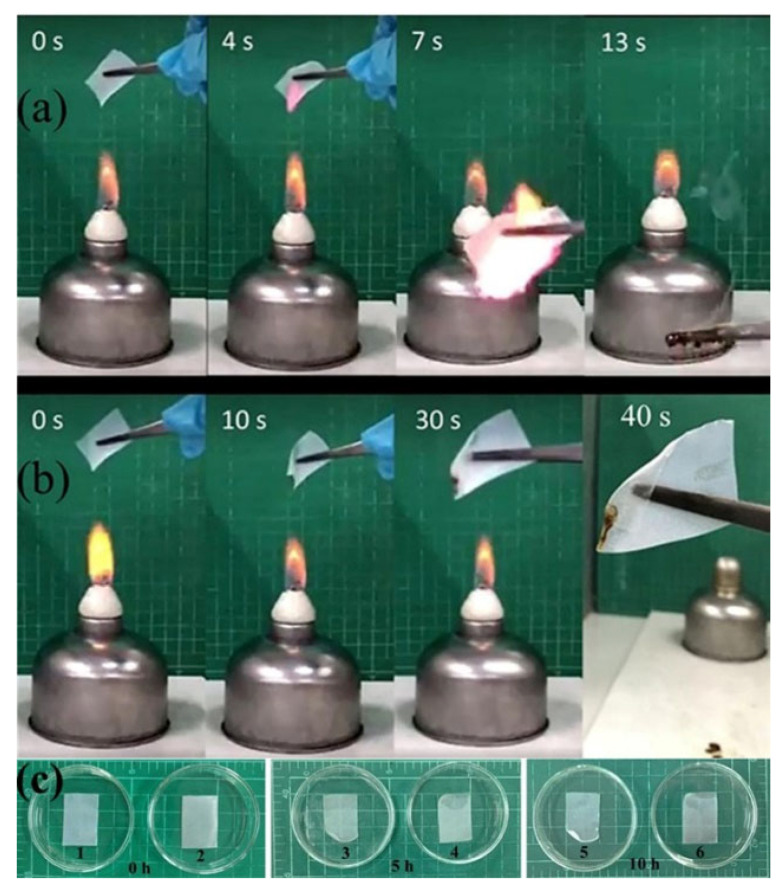
Flammability test of (**a**) PEO−20%LiCF_3_SO_3_−5%HNT, (**b**) PEO−20%LiCF_3_SO_3_−5%HNT−15 US. (**c**) dimensional stability at 60 °C of (1), (3), (5) for PEO−20%LiCF_3_SO_3_−5%HNT at 0 h, 5 h and 10 h, and (2), (4), (6) for PEO−20%LiCF_3_SO_3_−5%HNT−15 US at 0 h, 5 h, 10h.

**Figure 11 polymers-14-03710-f011:**
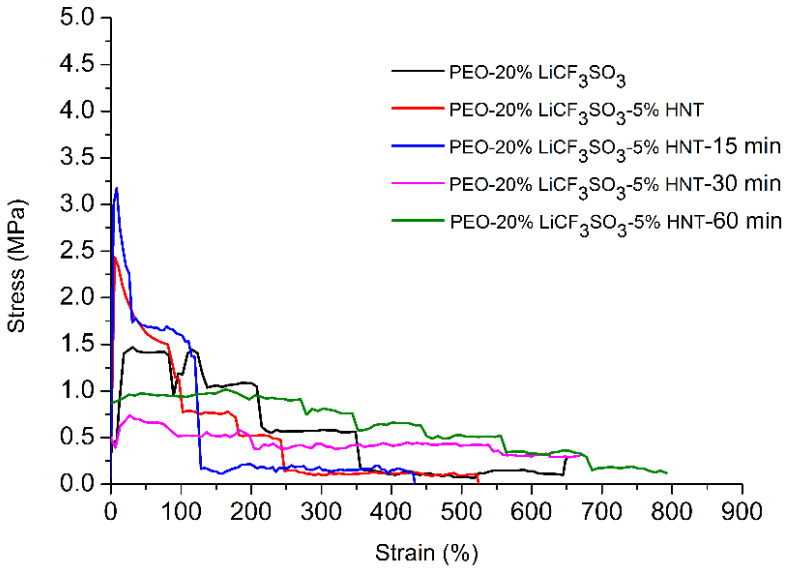
Stress–strain curves of solid polymer electrolytes at room temperature.

**Table 1 polymers-14-03710-t001:** T_g_, T_m_, ΔH_m_, and degree of crystallinity of SPE samples.

Samples	T_g end point_ (°C)	T_m_ (°C)	ΔH_m_ (J/g)	Degree of Crystallinity (%)	
				DSC	XRD
PEO−20%LiCF_3_SO_3_	−60.7	58.2	67.32	31.50	35.25
PEO−20%LiCF_3_SO_3_−5%HNT	−67.9	58.4	54.09	23.41	30.26
PEO−20%LiCF_3_SO_3_−5%HNT−15 min	−67.7	58.8	52.54	22.74	26.24
PEO−20%LiCF_3_SO_3_−5%HNT−30 min	−67.8	58.7	68.38	29.60	35.58
PEO−20%LiCF_3_SO_3_−5%HNT−60 min	−60.1	62.4	69.34	30.01	36.11

**Table 2 polymers-14-03710-t002:** The assignment of each wavenumber of FT-IR spectrum in Figure 8.

Samples	Wavenumber (cm^−1^)								
	Box1	Box2	Box3	Box4	Box5	Box6	Box7	Box8	Box9
	CF_3_ Stretch	CH_2_ Rock	CH_2_ Rock	C-O-C	CH_2_ Twist	CH_2_ Wag	CH_2_ Scissor	CH_2_ Stretch	OH Stretch
a	639	841	960, 948	1147, 1097, 1060	1279, 1259, 1241	1359, 1341	1467, 1454	2882	3494
b	639	841	961, 946	1147, 1097, 1060	1279, 1259, 1242	1359, 1342	1467, 1455	2884	3487
c	641	844	963, 949	1148, 1098, 1062	1281, 1260, 1244	1361, 1343	1469, 1457	2885	3493
d	638	842	960, 947	1146, 1096, 1061	1278, 1255, 1242	1359, 1341	1467, 1454	2881	3487
e	638	842	960, 946	1144, 1095, 1061	1278, 1253, 1266	1359, 1342	1467, 1455	2882	3487

**Table 3 polymers-14-03710-t003:** The ionic conductivity at room temperature of solid polymer electrolytes.

Solid Polymer Electrolytes	Ultrasonication Time (min)	Ionic Conductivity (S/cm)
PEO−20%LiCF_3_SO_3_−5%HNT	0	6.6 ± 1.33 × 10^−6^
	5	1.3 ± 0.83 × 10^−5^
	10	1.5 ± 0.03 × 10^−5^
	15	1.1 ± 0.19 × 10^−4^
	30	1.7 ± 0.84 × 10^−5^
	60	1.5 ± 1.24 × 10^−5^
PEO−20%LiCF_3_SO_3_−10%HNT	0	1.4 ± 0.17 × 10^−6^
	15	5.1 ± 1.15 × 10^−5^
	30	3.6 ± 1.15 × 10^−5^

**Table 4 polymers-14-03710-t004:** Tensile properties of SPEs.

Samples	Tensile Strength (MPa)	%Strain at Break	Stress @ 2% Offset of Strain
PEO-20%LiCF_3_SO_3_	1.47 ± 0.99	793.54 ± 3.46	0.43
PEO-20%LiCF_3_SO_3_-5%HNT	2.43 ± 0.44	525.01 ± 44.72	1.04
PEO-20%LiCF_3_SO_3_-5%HNT-15 min	3.18 ± 0.06	433.34 ± 57.74	1.53
PEO-20%LiCF_3_SO_3_-5%HNT-30 min	0.71 ± 0.12	799.98 ± 0.21	0.40
PEO-20%LiCF_3_SO_3_-5%HNT-60 min	1.02 ± 0.22	791.87 ± 4.33	0.87

## Data Availability

Data are available upon request from the corresponding author.

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
