# Peer review of "Effect of Ultrasonication on the Morphology, Mechanical Property, Ionic Conductivity, and Flame Retardancy of PEO-LiCF3SO3-Halloysite Nanotube Composites for Use as Solid Polymer Electrolyte"

_polymers, 2022, doi:10.3390/polym14183710_

Round 1
Reviewer 1 Report
This work involves PEO based solid polymer electrolyte. The authors studied the effects of ultrasonic treatment on the dispersion of nanofillers, matrix morphology, ionic conductivity, electrochemical stability, flame retardance and dimensional stability, etc. Some of the results are interesting, but there are some severe flaws in the methodology, data analysis and quality of manuscript preparation.
1. Line 43: the authors stated that the target ionic conductivity is 10^-5 S/cm for commercial LIBs but didn’t give reference. However, for this requirement, >10^-4 S/cm had been stated by many researchers. Please explain;
2. Line 53-55: No reference for the statement;
3. Line 87-88: what was the solvent used?
4. Line 112-116: DSC measurement: Usually the start and final temperatures should be far enough below or above the transitions to establish a stable baseline. Generally, the baseline should have two (2) minutes to completely stabilize prior to the transition of interest. Therefore, at 10°C/min., start at least 20°C below the transition onset temperature. In this work, the reported Tg was close to -70C, please explain how to ensure the results are reliable?
5. Line 115: Tg – onset or midpoint?
6. Line 207-211: The multi-peaks: why and what are the implications?
7. Line 237-239: PEO chain scission: Any evidence or reference?
8. Table 3: the ionic conductivity for Non-US of PEO-20%LiCF3SO3-10%HNT is not consistent with Figure 9b;
9. Many opportunities of improvement in manuscript preparation. Examples: Line 29 – a comma should be added before “the use of…”, Line 152 – “tend to be more prone to…” seems repetitive.
Author Response
Responses to Reviewer 1 (Round 1)
This work involves PEO based solid polymer electrolyte. The authors studied the effects of ultrasonic treatment on the dispersion of nanofillers, matrix morphology, ionic conductivity, electrochemical stability, flame retardance and dimensional stability, etc. Some of the results are interesting, but there are some severe flaws in the methodology, data analysis and quality of manuscript preparation.
Thank you so much for your value suggestion.
- Line 43: the authors stated that the target ionic conductivity is 10^-5 S/cm for commercial LIBs but didn’t give reference. However, for this requirement, >10^-4 S/cm had been stated by many researchers. Please explain;
The minimum requirements of ionic conductivity for all solid-state polymer electrolytes in LIBs is 10^-5 S/cm. It is stated in the reference below. However, I agree with the reviewer that the practical requirement needs to be above ≥10^−4 S cm−1; therefore, I edited the manuscript.
Reference: L. Yue, J. Ma, J. Zhang, J. Zhao, S. Dong, Z. Liu, G. Cui, L. Chen, All solid-state polymer electrolytes for high-performance lithium ion batteries, Energy Storage Mater. 5 (2016) 139-164.
- Line 53-55: No reference for the statement;
The references are added in the revised manuscript.
- Line 87-88: what was the solvent used?
THF was used as solvent, and it is stated in the revised manuscript.
- Line 112-116: DSC measurement: Usually the start and final temperatures should be far enough below or the transitions to establish a stable baseline. Generally, the baseline should have two (2) minutes to completely stabilize prior to the transition of interest. Therefore, at 10°C/min., start at least 20°C below the transition onset temperature. In this work, the reported Tg was close to -70C, please explain how to ensure the results are reliable?
The cooling system of this DSC apparatus cannot go down below -70 °C. However, the Tg is clearly seen on the expanded part of the DSC thermogram. Please see the below figure
- Line 115: Tg – onset or midpoint?
The Tg is obtained from the midpoint. Please see the above figure.
- Line 207-211: The multi-peaks: why and what are the implications?
PEO has a monoclinic crystalline structure with its (120) planes parallel to the PEO chain direction and (112) plane [reference is shown below]. When the LiCF3SO3, HNT and the ultrasonication were applied to PEO solution, they may interfere the crystal structure somehow, so the multi-peak (or the peak tends to split) is observed. However, the 2theta position does not change, this means the crystal form should be the monoclinic form as the neat PEO.
Reference: X. Xu, L, Jiang, Z. Zhou, X. Wu, Y. Wang, Preparation and Properties of Electrospun Soy Protein Isolate/Polyethylene Oxide Nanofiber Membranes. ACS Appl. Mater. Interfaces 4 (2012) 4331−4337.
- Line 237-239: PEO chain scission: Any evidence or reference?
The references are added.
- Table 3: the ionic conductivity for Non-US of PEO-20%LiCF3SO3-10%HNT is not consistent with Figure 9b;
Please accept my apology. Figure 9b in the revised manuscript is edited.
- Many opportunities of improvement in manuscript preparation. Examples: Line 29 – a comma should be added before “the use of…”, Line 152 – “tend to be more prone to…” seems repetitive.
The revised manuscript was proofread and edited by the native speaker.

Reviewer 2 Report
In this paper, the authors documents and discusses research work associated with a contextual overview of effect of ultrasonication on the morphology, mechanical prop- erty, ionic conductivity, and flame retardancy of PEOLiCF3SO3-Halloysite nanotube composites for use as solid polymer electrolyte. However, certain concerns need to be addressed:
1. The paper should add some figures appropriately.
2. Does the duration of ultrasonication have any effect on the mechanical properties, ionic conductivity, and flame retardancy of the composites?
3. Figure 10 and figure 11 can be combined into one group figure.
4. The conclusion is too short and must be improved, highlighting the purpose of the paper, summarizing the methods used and the results achieved.
5. There are some grammar and word mistakes in the manuscript. Please go through the manuscript carefully again.
Author Response
Response to Reviewer 2 Comments
(Round 1)
Comments and Suggestions for Authors
In this paper, the authors documents and discusses research work associated with a contextual overview of effect of ultrasonication on the morphology, mechanical property, ionic conductivity, and flame retardancy of PEOLiCF3SO3-Halloysite nanotube composites for use as solid polymer electrolyte. However, certain concerns need to be addressed:
Thank you so much for your value suggestion.
- The paper should add some figures appropriately.
Figures were modified and edited. Figure 7 is modified to see Tg better, and Figure 9 is edited.
- Does the duration of ultrasonication have any effect on the mechanical properties, ionic conductivity, and flame retardancy of the composites?
The results showed that the ultrasonication affects the ionic conductivity as shown in the manuscript; however, the effect on mechanical property and flame retardancy have not been explored in this work.
- Figure 10 and figure 11 can be combined into one group figure.
Figure 10 and 11 were combined into one group figure as shown in the new Figure 10.
- The conclusion is too short and must be improved, highlighting the purpose of the paper, summarizing the methods used and the results achieved.
The conclusion is edited.
- There are some grammar and word mistakes in the manuscript. Please go through the manuscript carefully again.
The revised manuscript was proofread and edited by the native speaker.

Round 2
Reviewer 1 Report
1. DSC Tg: As the author explained, the Tgs were midpoints. However from the thermograms, I cannot see how the two parallel lines can be drawn since the onset points are obviously below -70C. I would the authors redo the test with an appropriate equipment.
2. Table3/Figure 9b: I cannot see the differences after editing. Figure 9b in the revised version looks like the same as that in the original manuscript. The issue I brought up is the ionic conductivity of Non-US of PEO-20%LiCF3SO3-10%HNT. in Table 3, the reading is 1.412+/-0.17X10^-6 S/cm, while in Figure 9b, it looks close to 5X10^-6 S/cm.
Author Response
Thank you so much for your comment. I have learned a lot from it. However, I could not access the DSC which has the better performance at time time, so I have tried to use the results that I have in a better way. Please see the attach file for the answer.

Round 3
Reviewer 1 Report
Revision accepted.